# FFCL: Forward-Forward Contrastive Learning for Improved Medical Image Classification

**Md. Atik Ahamed**                                        ATIKAHAMED@UKY.EDU

**Jin Chen**                                                    CHEN.JIN@UKY.EDU

**Abdullah-Al-Zubaer Imran**                               AIMRAN@UKY.EDU

*University of Kentucky, Lexington, KY, USA*

## Abstract

Medical image classification is one of the most important tasks for computer-aided diagnosis. Deep learning models, particularly convolutional neural networks, have been successfully used for disease classification from medical images, facilitated by automated feature learning. However, the diverse imaging modalities and clinical pathology make it challenging to construct generalized and robust classifications. Towards improving the model performance, we propose a novel pretraining approach, namely **Forward Forward Contrastive Learning (FFCL)**, which leverages the Forward-Forward Algorithm in a contrastive learning framework–both locally and globally. Our experimental results on the chest X-ray dataset indicate that the proposed FFCL achieves superior performance (**3.69%** accuracy over ImageNet pretrained ResNet-18) over existing pretraining models in the pneumonia classification task. Moreover, extensive ablation experiments support the particular local and global contrastive pretraining design in FFCL.

**Keywords:** CNN, Forward-Forward Algorithm, Back-propagation, Chest X-ray, Pneumonia

## 1. Introduction

The imperative for automated disease diagnosis is contingent upon the accurate classification of medical images. In this context, deep convolutional neural networks (CNN) have proven to be remarkably effective in performing medical image classification tasks, thereby significantly contributing to the advancement of the automated diagnosis process. Nevertheless, CNNs exhibit limitations in terms of generalizability and the ability to capture fine details within input images. To address these limitations, we propose a multistage pretraining method with back-propagation, which effectively improves the performance of medical image classification and enhances model generalizability. According to the forward-forward algorithm (FFA) in Hinton (2022), there is no convincing evidence to suggest that our brain stores gradients and undergoes learning via a back-propagation mechanism. Moreover, the majority of contemporary disease classification tasks are carried out by training state-of-the-art deep learning models using the back-propagation approach. These models were typically trained from scratch using medical images or fine-tuned based on the ImageNet-pretrained models (Deng et al., 2009). However, ImageNet does not well represent the characteristics of images within the medical imaging domain, resulting in suboptimal model generalizability. In this project, we leverage supervised contrastive learning (Khosla et al., 2020) as a pretraining strategy, instead of directly utilizing back-propagation with weights pretrained on the ImageNet dataset. Existing studies demonstrate contrastive learning performed by only taking the final output of the model, which does not capture the fine image details. In this work, we

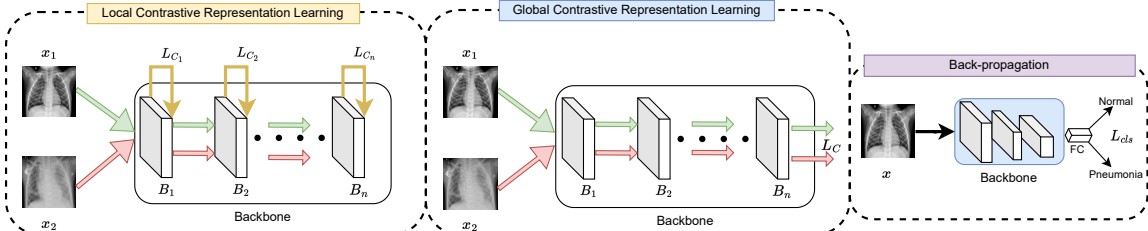

Figure 1: The FFCL model and the multistage contrastive pretraining strategy. $L_{C_i}$ & $L_C$ represent local & global contrastive loss, respectively. $B_i, i = [1, n]$ represents a block of the target model.

propose a multistage pretraining strategy, called FFCL, before performing back-propagation. We pretrain our backbone model in a supervised contrastive representation learning manner locally for each layer, and globally for the target model to capture fine details of the input image. Our pretraining strategy is inspired by the forward-forward algorithm (FFA). However, our solution does not require manual fine-tuning of any thresholds, which helps the model to capture local information automatically and reduce manual effort. To the best of our knowledge, this is the first work leveraging contrastive learning in a forward-forward mechanism in medical imaging. Notably, the proposed FFCL can be extended further to train a model end-to-end without requiring any manual intervention in between.

## 2. Methods

FFCL comprises two stages of pretraining before performing regular back-propagation for the downstream classification tasks. Fig. 1 shows the FFCL framework, which comprises two pretraining stages (local and global contrastive representation learning) and the final back-propagation-based image classification. In the first stage, we perform contrastive learning locally, based on the modified Forward-Forward Algorithm (FFA) Hinton (2022). Unlike FFA, we leverage supervised contrastive learning for local updates at each layer of the target model without requiring tuning any thresholds. In the first stage, FFCL randomly takes two images $x_1, x_2 \in X_{train}$ sampled from the train set ($X_{train}$) and provides embeddings $E_{x_{1_i}}, E_{x_{2_i}}$ from each block $B_i$ followed by ReLU (Agarap, 2018) activation layer. We perform local updates utilizing cosine embedding loss $L_{C_i}$ for each of the blocks ($B_i$s). After performing local updates for each block, the first stage pretrained model is used for performing the global contrastive learning in the second stage. As in local contrastive learning, this global contrastive learning takes two random images as input and maps to the final embedding space ($E_{x_1}, E_{x_2}$). The same cosine embedding loss is used for global contrastive learning (see Eq. (1)). In the third stage, the latest pretrained model is leveraged for performing the actual downstream classification task with regular back-propagation. For both baseline and FFCL, we use the binary-cross-entropy loss for the downstream classification task. FFCL is elegant in the sense all the training stages are performed in a fully automated manner without requiring any hyper-parameter tuning in between.

$$L_c(E_{x_1}, E_{x_2}, C_{x_1}, C_{x_2}) = \begin{cases} 1 - \frac{E_{x_1} \cdot E_{x_2}}{\|E_{x_1}\|_2 \|E_{x_2}\|_2}, & \text{if } C_{x_1} = C_{x_2} \\ \max\left(0, \frac{E_{x_1} \cdot E_{x_2}}{\|E_{x_1}\|_2 \|E_{x_2}\|_2}\right), & \text{if } C_{x_1} \neq C_{x_2} \end{cases} \tag{1}$$

Table 1: Pneumonia classification performance with the ResNet-18 and ResNet-34 backbone networks. We compare FFCL against regular backpropagation (RBP), with random and ImageNet pretrained weight initializations. The ablation study is reported on FFCL using ResNet-18.

| Backbone | Approach | Contrastive | Initialization | Accuracy | F1 | Precision | Recall | AUC |
|---|---|---|---|---|---|---|---|---|
| ResNet-18 | RBP | – | ImageNet | 76.76 | 69.84 | 85.94 | 69.10 | 92.90 |
| | | – | random | 73.08 | 63.14 | 84.95 | 64.10 | 89.15 |
| | FFCL | Local→Global | ImageNet | 72.28 | 61.60 | 84.64 | 63.03 | 84.21 |
| | | Local→Global | random | **80.45** | **75.69** | **87.70** | **74.02** | **93.33** |
| | | Global→Local | random | 74.04 | 65.38 | 83.52 | 65.64 | 91.23 |
| | | Global only | ImageNet | 69.71 | 56.38 | 83.68 | 59.62 | 87.88 |
| | | Local only | ImageNet | 74.36 | 65.52 | 85.45 | 65.81 | 93.09 |
| | | Global only | random | 76.92 | 70.22 | 85.54 | 69.40 | 92.25 |
| | | Local only | random | 71.63 | 60.53 | 83.58 | 62.26 | 89.00 |
| ResNet-34 | RBP | – | ImageNet | 78.53 | 72.71 | **86.77** | 71.45 | 94.09 |
| | FFCL | Local→Global | random | **78.85** | **73.32** | 86.51 | **71.97** | **94.43** |

where, $(E_{x_1}, C_{x_1})$ & $(E_{x_2}, C_{x_2})$ denotes the embedding space and class label of the corresponding input image $x_1$ & $x_2$.

## 3. Experiments and Results

We perform binary classification (pneumonia vs. normal) from chest X-ray images employing the ResNet-18 and ResNet-34 backbones (He et al., 2016). We evaluated the proposed method using the pediatric chest X-ray dataset Kermany et al. (2018). The dataset contains a total of 5856 chest X-ray images split into train (5232) and test (624). For our experiments, we further separated 262 images from the train set as the validation set which was used for selecting the best models. All the images were resized to $224 \times 224 \times 3$ and 0–1 normalized (training from scratch) and ImageNet normalized (training from ImageNet weight) before passing them to the models. The models were trained with a Cosine Annealing scheduler (initial learning rate of 0.0001) using the Adam optimizer. We used a batch size of 10 and trained for 100 epochs for all the models for downstream classification task.

Table 1 compares the classification performance of our proposed approach against the baselines. For ResNet-18 local only from scratch, one sample was always from the normal class during forward-forward pretraining. For simplicity, we refer the regular backpropagation as RBP. Except accuracy and ROC-AUC, other reported metrics have been *macro* averaged. Table 1 demonstrates superior performance over the baseline RBP with an improvement of 3.69% in terms of accuracy. Moreover, our proposed FFCL method outperforms state-of-the-art pneumonia classification methods: AUC of 77% Jaiswal et al. (2021), AUC of 78.4% Seyyed-Kalantari et al. (2020) and AUC 75% Liu et al. (2019).

## 4. Conclusions

We proposed a novel multistage contrastive pretraining strategy (FFCL) to enhance disease detection with state-of-the-art deep learning models. Our proposed FFCL-based pretraining fully exploits the deep learning models' learnability by performing local and global updates. Our extensive experimentation along with innovative ablation study confirmed the superiority of FFCL over regular backpropagation training in performing pneumonia disease classification.

Our ongoing efforts include evaluating on larger-scale datasets of varying diseases as well as additional medical image analysis tasks.

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
