# OpenReview forum: "FFCL: Forward-Forward Contrastive Learning for Improved Medical Image Classification"
_MIDL.io/2023/Short_Paper_Track — MIDL 2023 Short paper track Poster_

### Official Review · Reviewer_PjvL · 2023-04-10
**This is a research paper on improving medical image classification using a pre-training approach called Forward Forward Contrastive Learning (FFCL).**

**Rating:** 8
**Confidence:** 4

**Review:**

This is a research paper on improving medical image classification using a pre-training approach called Forward Forward Contrastive Learning (FFCL). The authors propose a multistage pre-training method with back-propagation to enhance the performance of medical image classification and model generalizability. They leverage supervised contrastive learning in a forward-forward mechanism in medical imaging and perform binary classification from chest X-ray images. The experimental results show that the proposed FFCL achieves superior performance over existing pre-training models in the pneumonia classification task. The proposed method is automated, does not require any manual fine-tuning of thresholds, and can be extended to train a model end-to-end.

Pros:
The proposed Forward Forward Contrastive Learning (FFCL) approach achieves superior performance over existing pretraining models in the pneumonia classification task.
The FFCL approach leverages supervised contrastive learning for local updates at each layer of the target model, capturing fine details of input images.
FFCL is fully automated, requiring no manual intervention in between the training stages, making it an elegant and efficient pretraining method.
The proposed method can be extended further to train a model end-to-end without requiring any manual intervention in between.
The study shows the limitations of using ImageNet-pretrained models for medical image classification, as ImageNet does not represent the characteristics of images within the medical imaging domain.

Cons:
The study is limited to chest X-ray images and may not be generalizable to other imaging modalities or clinical pathology.
The study does not compare the proposed FFCL approach with other state-of-the-art pretraining methods, making it unclear how it compares to other approaches in the field.
The study does not provide insights into how the proposed method might perform on larger datasets or with different architectures.

---

### Official Review · Reviewer_APEe · 2023-04-24
**a contrastive learning method based on the the FF algorithm to improve disease classification performance on X-ray images**

**Rating:** 7
**Confidence:** 4

**Review:**

The authors propose the use of a contrastive learning strategy based on the forward-forward algorithm by Hinton 2022, entitled “Forward-Forward Contrastive Learning” which is applied to pretrain a model to perform pneumonia detection in pediatric X-ray images.

I believe the method is interesting and could have potential as a contrastive learning strategy to improve performance on image classification. The experimental setup is a bit weak in the sense that it only includes a single dataset which is pretty small for this type of problem (around 5000 images, while other datasets like CheXPert or NIH contain more than 100.000 images). However, since the proposed strategy is timely and seem to work, I think it could spark interesting discussions at MIDL around the topic of contrastive learning.

The model was evaluated on the ResNet-18 and 34 backbones, and it seems to result in better improvement for the smaller model (ResNet-18). This could imply that the proposed strategy can help specially in case of smaller models; however, this should be further investigated and support by additional experiments. I recommend the authors to continue their work on this direction.